# Associations between Stressful Life Events and Increased Physical and Psychological Health Risks in Adolescents: A Longitudinal Study

**DOI:** 10.3390/ijerph20021050

**Published:** 2023-01-06

**Authors:** Anna Roth, Christof Meigen, Andreas Hiemisch, Wieland Kiess, Tanja Poulain

**Affiliations:** 1LIFE Leipzig Research Center for Civilization Diseases, Leipzig University, Philipp-Rosenthal-Strasse 27, 04103 Leipzig, Germany; 2Department of Women and Child Health, University Hospital for Children and Adolescents and Center for Pediatric Research, Leipzig University, Liebigstrasse 20a, 04103 Leipzig, Germany

**Keywords:** stressful life events, adolescence, health related quality of life, physical health, psychological health, socioeconomic status, sex, age

## Abstract

Stressful life events (SLEs) are understood as risk factors for mental and physical health problems, particularly in the vulnerable period of adolescence. Using a longitudinal approach, this study investigated associations between SLE and several negative health outcomes in adolescents. Moderating effects of sociodemographic factors were considered. We analyzed the data of a healthy adolescent sample from the LIFE Child study in Leipzig, Germany (*n* = 2024, aged 10–18 years). SLEs were measured by a questionnaire, addressing SLEs in the family and the social environment domain. Health-related quality of life (HrQoL), behavioral difficulties and BMI were compared before and after an SLE had occurred. Moderator effects of socioeconomic status (SES), age, and sex were investigated using linear regression models. All considered health parameters had, on average, deteriorated after the occurrence of an SLE in the social environment. Differences in HrQoL before and after an SLE were significantly stronger in girls. Higher SES functioned as a slight protective factor against decreased well-being after an SLE. The findings suggest that SLEs function as risk factors for mental and physical health disadvantages in adolescents. Prevention programs should seek to support adolescents in all age and SES groups affected by SLEs, with a specific focus on girls.

## 1. Introduction

Within the last decades, many investigations on the effects of life stress on human health and well-being have shown associations between psychological stress and mental and physical health risks such as depression, cardiovascular disease, and premature mortality [1,2]. Chronic stress exposes an organism to higher levels of stress hormones and an activation of the hypothalamic–pituitary–adrenergic (HPA) axis that exceeds the physiological adaptation to short-term stress situations [3,4]. Stressful life events (SLEs) are stressors that require an adaptation to a new situation [5] and exceed one’s resources [6].

SLEs during childhood or adolescence are prevalent; a U.S. American survey [7] revealed that 25% of a large representative sample had experienced a high magnitude SLEs such as being a victim of physical violence before the age of 16. One third of the respondents reported a low magnitude SLE such as the death of a grandparent within the last three months. In Germany, 43.7% of adults had undergone at least one SLE during childhood, with parental separation and divorce being most frequent [8].

Long term effects of SLEs in early childhood, e.g., abuse or neglect, are increasingly understood as particularly detrimental [9,10,11,12]. Examples of potential negative effects include obesity and overweight [11], early-onset depression [12], and increased suicidality [10]. Often, the consequences do not become apparent until adulthood [9].

In contrast, SLEs during adolescence, a vulnerable phase characterized by many hormonal, psychological, and behavioral changes [13,14], remain rather underrepresented in the current research. The period of adolescence is particularly vulnerable for stressors that occur in addition to these existing simultaneous changes, which may exceed the ability to cope [13].

Several longitudinal and cross-sectional studies have revealed SLEs during adolescence to be risk factors for negative health outcomes; at the level of psychological problems, Baddam et al. found significant associations of recent stress events with sleep disturbances in 752 12- to 15-year-old adolescents [15]. Additionally, SLEs appear to be related to internalizing symptoms such as depressive mood [16] and anxiety, mediated by emotional dysregulation [17]. Yildiz et al. found that SLEs are associated with increased suicidal ideation and attempts [18]. Song et al. found significant associations between SLEs and behavioral problems in more than 5000 Chinese third- to eighth-graders, using a cross-sectional study design [19]. The health-related quality of life (HrQoL), a marker of an individual’s well-being, is also affected by SLE in adolescence [20,21,22]. Overall, two large meta-analyses summarized the detrimental effects of SLEs on mental health and behavior [23,24]. At the level of physical problems, nicotine and drug consumption increased after SLEs in U.S. American adolescent samples [16,25]. In a Canadian retrospective study, 13- to 17-year-old male adolescents reporting one recent event were nearly 50% more likely to have obesity at 6-month follow-up compared to young men reporting no SLE [26]. Furthermore, SLEs during adolescence are associated with physical health characteristics such as changes in white matter properties in young men’s brains [27] or in cardiometabolic biomarkers [28].

Some of the mentioned studies follow cross-sectional models that can only cover the very immediate effects of SLEs [16,19,20,26]. Others only address SLEs from a specific life domain such as problems within the family context [20,21]. Likewise, some studies only analyze a small age range [15,17,18,20] or examine associations between SLEs and a specific outcome variable. Most do not consider the possible influencing factors that may mediate or moderate the associations between SLEs and health parameters, e.g., socioeconomic status (SES) or age. Sex differences are highlighted by a few studies, indicating somewhat stronger associations in females [16,26].

Addressing some of these research gaps, this paper targeted possible associations between SLEs in adolescence and negative health outcomes using a longitudinal design. Specifically, we compared health parameters before and after the occurrence of SLEs in a large adolescent sample, generating a more precise idea of stress effects in adolescents. Contrasting a time point when adolescents did not report an SLE with a time point after the occurrence of an SLE represents the novelty of this study. Furthermore, we analyzed SLEs in different life domains, i.e., the family and the social environment, in a large population-based sample and investigated several health parameters from the fields of psychological, somatic, and behavioral problems. Finally, we assessed whether associations between the experience of an SLE and health were moderated by the sociodemographic factors of age, sex, and SES. Previous studies showed that behavioral difficulties and the quality of life of children differ between girls and boys, between different age groups, and between different social strata [29,30,31]. Therefore, it is to be expected that associations between the experience of SLE and health will also differ between these groups.

Based on the reviewed literature, we hypothesized that SLEs in adolescents aged 10 to 18 years are associated with health disadvantages, such as a higher body mass index (BMI), poorer HrQoL, and more behavioral difficulties. Furthermore, we hypothesized that these associations are stronger in girls and in adolescents from families with a lower SES.

## 2. Materials and Methods

### 2.1. Participants/Study Sample

The present study was performed within the LIFE Child study, conducted at the Leipzig Research Center for Civilization Diseases at Leipzig University, Germany. It aimed to investigate the influencing factors on children’s development and the etiology of non-communicable diseases, with a special focus on mental health problems and obesity [32,33]. The LIFE Child study follows healthy subjects from the prenatal period to early adulthood in optimally annual follow-up examinations. Participants mainly come from the urban and rural areas of Leipzig, a city in Eastern Germany with more than 600,000 inhabitants (as for 2021). Compared to other cities, especially west German cities, Leipzig is a city with a relatively high poverty and unemployment rate [34]. Nevertheless, Leipzig has been reported to be a city with a high living quality [35]. Informed written consent for the study program is obtained from all parents and child participants aged 12 years and older. The study is conducted in accordance with the Declaration of Helsinki and was approved by the Ethics Committee of the University of Leipzig (Reg. No. 264/10-ek). Patients or the public were not involved in the design, conduct, reporting, or dissemination plans of our research.

All participants who had completed the Life Events Questionnaire at at least one time point between 2011 and 2021 were eligible for the present analyses (*n* = 2024 10- to 18-year-old adolescents). Of these children, 36.4% had participated at one time point, 40.6% at two to four time points, and 22.9% at more than four time points. The *descriptive analysis*, i.e., analyses regarding the frequency and number of experienced SLEs, was performed in these 2024 individuals. For the *longitudinal analysis*, only subjects who had participated at two subsequent study visits and who had experienced at least one SLE were eligible. Those who reported having not witnessed any of the queried SLEs at any time (*n* = 572) and those for whom no long-term data were available (*n* = 737) were excluded, resulting in 715 individuals eligible for the longitudinal analysis. For visualization of participant selection see Figure 1.

### 2.2. Life Events Questionnaire

In the LIFE Child study, the occurrence of an SLE in the last six months is recorded by means of a self-created questionnaire completed by the participants at each study visit [33]. A wide variety of possible life events are queried. The response options are “yes, I have experienced this SLE in the last six months” and “no, I have not experienced this SLE in the last six months”.

The following SLEs were included in the present analyses: “divorce of parents”, “death of a close family member”, and “unemployment within the family” (categorized as “family problems” (SLE-fam)), as well as “victim of bullying or violence”, “problems among friends”, “moving to a different place of residence”, and “separation from partner” (categorized as “problems in the social environment” (SLE-soc)). A participant was considered to have experienced an SLE-fam or SLE-soc if at least one of the corresponding single SLEs had occurred within the last six months.

### 2.3. KIDSCREEN-27

The subjective health-related quality of life (HrQoL) was analyzed by means of the KIDSCREEN-27 questionnaire (self-report). This instrument covers five dimensions of HrQoL: *physical well-being, psychological well-being, parent relations and autonomy, social support and peers, and school environment* [36].

For evaluation, age- and sex-specific t-values (mean = 50, sd = 10) were used. Since this study focused especially on well-being and (mental) health, we only considered the scales *physical well-being* and *psychological well-being* as outcome variables. In a large study in 28 European countries, internal consistencies of these scales (indicated by Cronbach’s alpha) were high (α = 0.80 and 0.84, respectively) [37]. In the current sample, the internal consistencies were even higher (α = 0.81 and 0.86, respectively). The validity of the instrument was shown elsewhere [31].

### 2.4. Strengths and Difficulties Questionnaire (SDQ)

The SDQ (self-report) was applied to investigate children’s and adolescents’ strengths and difficulties. It comprises twenty-five items assessing five different dimensions, namely the behavioral difficulties *conduct problems*, *emotional symptoms*, *hyperactivity/inattention*, and *peer relationship problems*, as well as the behavioral strength *prosocial behavior*. The sum scores of the difficulties scales could be combined to a total difficulties score ranging from 0 to 40, with higher scores indicating more behavioral difficulties [38]. This score was included as the outcome variable.

In the present sample, the internal consistency of the total difficulties score was acceptable (α = 0.76), which was in line with findings in a large German survey (α = 0.72) [29]. The validity of the SDQ was shown in previous studies [39].

### 2.5. Body Mass Index (BMI)

The weight and height of children were measured by trained study assistants. The BMI values were transformed to standard deviation scores (BMI-SDS) based on percentile curves of age- and sex-specific German reference samples [40].

### 2.6. Socioeconomic Status (SES)

Children’s socioeconomic status (SES) was assessed using an SES composite score (adapted to the “Winkler–Stolzenberg Index” [41]), based on parent-provided information on their education, occupational status, and family net income. The index can take values ranging from 3 to 21 points, with higher values implicating a higher SES [41].

In accordance with a large national survey on child health in Germany [42], the values of the composite score can be used to classify a family’s SES as low (3–8.4 points), middle (8.5–15.4 points), and high (15.5–21 points). If a sample includes 20% from the low, 60% from the middle, and 20% from the high SES group, it can be considered representative [42].

### 2.7. Data Analysis

Statistical calculations and visualization were performed using R version 4.0.5. [43].

In the *descriptive analysis* (*n* = 2024), all adolescents having reported an SLE-fam or an SLE-soc at at least one time point were considered as having experienced the SLE.

In the *longitudinal analysis* (*n* = 715), we compared health parameters at two time points, once before experiencing the SLE-fam or SLE-soc, and once after the SLE had occurred. We first identified the time point at which the participant first reported to have experienced a specific SLE within the last six months. Thus, this time point could vary from participant to participant depending on the individual questionnaire report. This time point was referred to as “post-SLE” (post-SLE-fam or post-SLE-soc). We then determined the time point previous to the “post-SLE”-event, referred to as “pre-SLE” (pre-SLE-fam or pre-SLE-soc). For 92% of participants, the time span between “pre-SLE” and “post-SLE” was one year, meeting the annual follow-up objective of the LIFE Child study. For 7%, the difference was two years, and for 1% it was three or four years. For each subject, only the data points pre-SLE and post-SLE were considered. We then applied paired sample t-tests to explore the mean differences in the outcome variables between the groups (post-SLE-fam vs. pre-SLE-fam and post-SLE-soc vs. pre-SLE-soc). BMI-SDS, the SDQ total difficulties score, and the scores on the psychological and physical well-being scales of the KIDSCREEN-27 were included as dependent variables.

After identifying the relevant associations between an SLE and the considered health parameters, we investigated whether the magnitudes of differences in health parameters before and after an SLE differed depending on age, sex, or SES (*moderator analysis*). We applied multiple linear regression models with the differences in health parameters included as the dependent variables, and age, sex, and SES included as the independent variables.

For all the analyses, the significance level was set to α = 0.05. To control for multiple testing, the *p*-values were adjusted using the false discovery rate (FDR) method.

## 3. Results

### 3.1. Study Sample

The present sample comprised 2024 adolescents (1009 girls, 1015 boys) aged from 10 to 18 years (mean age = 13.09, sd = 2.19). Regarding SES, 11.5% belonged to the “low” SES group, 53.3% to the “middle” SES group, and 35.2% to the “high” SES group. The mean SES was 13.53 points (sd = 3.81). This showed a tendency towards higher social strata.

The mean SDS-BMI was 0.28 (sd = 1.22), implying a slightly higher average BMI value compared to the age- and sex-specific percentile curves of a German reference sample [40].

### 3.2. Descriptive Analysis

The frequencies of the different life events are shown in Table 1. Overall, 805 individuals (40.0%) reported having experienced at least one SLE-fam in the last six months. The death of a close relative was the SLE-fam that was experienced most frequently (*n* = 577, 29.0%). SLEs in the social domain were reported more frequently. A total of 1271 participants (64.1%) had experienced at least one SLE-soc. Being a victim of bullying or violence was the SLE-soc reported most frequently (*n* = 864, 42.8%).

The average number of different SLEs experienced by each subject per examination time point was 1.48 (sd = 1.72). The numbers of different SLEs were weighted by the inverse of the number of follow-ups of each participant, giving each participant equal weight.

### 3.3. Longitudinal Analysis: Differences in Health Parameters before and after Experiencing an SLE (Two Sample T-Tests)

Regarding SLE-fam, the t-tests revealed a significantly lower psychological well-being score after an SLE-fam (M = 49.43, sd = 9.81) than before an SLE-fam (M = 50.50, sd = 9.86) (*p* = 0.02). This implied a deterioration in psychological well-being. The mean SDS-BMI value was significantly higher after an SLE-fam (M = 0.34, sd = 1.24) than before an SLE-fam (M = 0.28, sd = 1.24), *p* < 0.01). The differences in the means of the other considered health parameters (KIDSCREEN-27 physical well-being score and SDQ total difficulties score) did not reach statistical significance.

Regarding SLE-soc, all health parameters were shown to differ significantly before and after an SLE-soc had occurred. While the mean KIDSCREEN-27-scores for physical well-being and psychological well-being were lower after an SLE-soc than before an SLE-soc (*p* < 0.01, respectively), the mean SDQ total difficulties score was higher after an SLE-soc (M = 9.92, sd = 4.77) than before (M =8.93, sd = 4.08) (*p* < 0.01). Similarly, the mean BMI-SDS value was significantly higher after an SLE-soc (M = 0.23, sd = 1.13) than before an SLE-soc (M = 0.19, sd = 1.11) (*p* = 0.036). These differences indicated an increase in health risks after an SLE-soc had occurred. All the t-test results are summarized in Table 2 and Figure 2.

### 3.4. Moderator Analysis: Associations between SES, Age and Sex, and Differences in Health Parameters (Linear Regression Analyses)

Regarding SLE-fam, the analyses revealed a significantly stronger decrease in the KIDSCREEN-27 scores for physical well-being and psychological well-being in girls than in boys (b = −2.53 and −2.74, *p* = 0.002 and 0.001, respectively). These results implied that the decrease in these well-being scores from pre-SLE-fam to post-SLE-fam was stronger in girls than in boys.

Regarding the SES as a possible moderating factor, the magnitude of the decrease in the physical well-being score was negatively associated with a higher SES (b = 0.28, *p* = 0.02). The analyses revealed no other significant interactions with sex, SES, or age.

Regarding SLE-soc, we observed similar results. As for SLE-fam, decreases in the physical well-being score as well as in the psychological well-being score after an SLE-soc compared to before an SLE-soc were significantly stronger in girls than in boys (b = −3.94 and −4.54, both *p* < 0.001). However, all other interactions with sex, SES, or age did not reach statistical significance.

Overall, the moderator analysis revealed that the female sex represented a risk factor for changes in well-being after an SLE-fam or an SLE-soc, while a high SES might represent a resilience factor, at least for SLEs in the family domain. The results of the linear regression analyses can be seen in Table 3.

## 4. Discussion

The aim of this study was to advance the understanding of the associations between SLEs and adverse health outcomes in adolescents. We compared HrQoL, behavioral difficulties, and BMI before and after the experience of an SLE in adolescents and considered differences in the strengths of changes depending on SES, age, and sex. We used a large, population-based dataset and a longitudinal study approach.

### 4.1. Main Findings

Most of the considered health parameters significantly deteriorated after the experience of an SLE. This was consistent with large meta-analyses that identified SLEs in adolescents as a risk factor for health from the majority of the existing literature [23,24]. This can be considered the main finding of our study. All effect sizes, however, were relatively small, indicating that, overall, adolescents are also able to adapt to challenging life events.

Regarding differences between environmental SLEs and familial SLEs, our results showed that all the observed health parameters had, on average, declined compared to before the occurrence of an environmental SLE (e.g., bullying, break-up, problems with friendships, and/or moving to a new place of residence). Contrarily, after the participants had experienced difficulties in the family, (e.g., divorce of the parents, unemployment, or death of a relative), there was a significant deterioration in psychological well-being and an increase in the mean BMI-SDS, but no change in physical well-being and behavioral difficulties. This suggested that SLEs in the family domain may represent a smaller burden for adolescents than SLEs in the social environment. In addition, certain protective factors, e.g., family cohesion, may play a more decisive role in familial SLEs against the onset of pathological processes.

### 4.2. Associations between SLEs and Psychological Health Risks

Considering the associations between SLEs and both HrQoL and behavioral difficulties, our findings were in line with previous studies examining these relationships [20,21,22,23]. Coker et al. and Martin-Guiterrez et al. found that an increased number of SLEs in the family surroundings is associated with lower general HrQoL [20,21]. Villalonga-Olives et al., who also used the KIDSCREEN questionnaire, found an association between SLEs and a deterioration in all KIDSCREEN dimensions, with the strongest associations in the psychological well-being score [22].

There are different approaches to explain why the self-perceived health status decreased and behavioral difficulties arose after experiencing an SLE. First, chronic stress, e.g., after experiencing an SLE, influences the neuroendocrine mechanisms of an organism. Stress-induced activation of the HPA axis can affect metabolic and biochemical processes and alter brain structures [13]. During puberty, when many changes take place in the brain and hormonal system, additional HPA axis activation can increase psychopathological processes. This is reflected in a high prevalence of mental disorders in adolescence [3] and could be an explanation for the lower psychological well-being and more behavioral difficulties after an SLE. Stress hormones, especially cortisol, are understood to be powerful modulators of the immune system. Elevated stress hormone levels are associated with pathophysiological processes [4] and may, therefore, impact self-perceived physical well-being.

Second, the experience of an SLE in adolescence has been specifically identified as a predictor for higher substance use [25], poorer sleep quality [15], brain changes [27], and modified physical health parameters [28]. All of these can lower self-perceived HrQoL and impair behavioral adaptation.

### 4.3. Associations between SLEs and Physical Health Risks

The mean SDS-BMI values were higher after the occurrence of an SLE in both considered life domains. Conklin et al. reported similar results, linking SLEs with obesity [26]. Possible reasons for this might be that, firstly, stress-related emotional and behavioral problems lead to “comfort eating”, less exercise, and sleep deprivation, which are associated with weight gain [44]. Secondly, the molecular mechanisms linking stress-induced increased cortisol levels and fat accumulation have been previously reported [45], which may also explain the increased BMI values.

### 4.4. SLE in Different Life Domains

The present analysis revealed that substantially more participants experienced at least one SLE in the social domain than in the family domain. Similar observations were made by two studies that also examined SLEs in adolescents [22,26].

The dominance of SLEs in the social domain emphasizes the change in the psychosocial environment taking place during adolescence. Linked to this is a detachment from family dependency, increased interaction with peers, and decreased interaction with adults [13], which highlights the importance of healthy peer relationships during adolescence. However, increased peer group interaction also carries the potential for more dysfunctional experiences.

### 4.5. Moderator Analyses

In contrast to previous findings [21], our moderation analyses revealed no significant age influence on the strengths of associations between SLEs and the considered health parameters. According to our data, the interval between 10 and 18 years of age can be assumed as a vulnerable period for stress-related negative health outcomes, regardless of the exact time point in which the SLE occurs.

Importantly, we found significantly stronger decreases in the HrQoL after an SLE in girls than in boys. However, there were no sex differences regarding behavioral difficulties and BMI. The reviewed literature is inconsistent regarding the moderating effect of sex. Some studies found a greater level of subjective and objective stress [15,24], as well as stronger associations between SLEs and internalizing problems, particularly depressive symptoms, in girls [6,16]. In contrast, the associations between SLEs and smoking [25], conduct problems [19], obesity [26], and general HrQoL [22] were shown to be stronger in boys. Other studies did not find any sex differences [17,23]. The ambiguity of the impact of sex may be due to various factors, including differences in the prevalence of SLE types and different coping styles [46]. In addition, gender roles and social desirability may influence response patterns, as females are more likely to report emotional symptoms that are socially recognized as “weak” [46]. Our results suggested that SLEs can be considered as risk factors for negative health outcomes in both girls and boys, with girls being slightly more in danger of suffering from lower physical and psychological well-being.

Regarding SES as a possible moderating factor, we found significantly less deteriorated physical well-being after an SLE in the family domain in participants with higher SES values. Therefore, a higher SES, related to stronger social support [47], may function as a protective factor in this relationship. Contrarily, all the other associations were not moderated by SES, implying that SLEs can affect adolescents from higher socioeconomic backgrounds almost as equally as their peers with a lower family SES. Our findings were partially in line with the, to our knowledge, only other study examining this moderating effect; Reiss et al. used data from a large population-based German sample and found no moderation effect of household income and parental unemployment on the association between an SLE and children’s mental health. Only a higher level of parental education was found to moderate this association [30].

### 4.6. Strengths and Limitations

The present study had several strengths, including a large sample size, the longitudinal design, and up-to-date data. Furthermore, we used established and validated instruments such as the SDQ, KIDSCREEN-27, and measured SDS-BMI values. We considered different dimensions of health as well as common SLEs from different life domains and analyzed the moderator effects of SES, age, and sex.

Nonetheless, there were several limitations. Conclusions regarding the overall population of adolescents can only be drawn with caution, as the present sample was sourced from only one large German city. Moreover, the participants’ SES showed a tendency towards higher social strata, implicating limited representativeness. A further limitation concerns the assessment of SLEs. The applied life events questionnaire asked for SLEs in the last six months only, while SLEs occurring in the previous seven to twelve months were not registered. To gain greater sample sizes for the statistical analyses, we considered various single SLEs summed up to SLE domains (SLE-fam and SLE-soc). Therefore, we could not make a conclusion regarding the effects of specific SLEs. The life events questionnaire as well as the target instruments, SDQ and KIDSCREEN, were applied in self-report versions only, representing the participants’ subjective perception. Objective parent- or teacher-provided data might be insightful for a deeper understanding of the effects of SLEs. A further limitation was that potential confounding and mediating factors, such as individual social support and coping strategies, were not considered due to limited data availability. Lastly, the longitudinal design covering two time points did not allow for the investigation of the long-term effects of SLEs in adolescents.

## 5. Conclusions

SLEs, especially in the social domain, are risk factors that are part of the everyday reality of many adolescents and that show detrimental effects on their quality of life, physical health, and psychosocial adaptation, especially in girls.

The results of our study emphasized the negative effect of SLEs on adolescent health. Even in our large healthy adolescent sample from relatively well-situated families, we identified potentially negative health outcomes related to SLEs. SLEs are therefore of relevance to public health. Parents, healthcare professionals and educators should identify and address SLEs as risk factors for their protégés’ health, regardless of their age or SES. Support and prevention measures should be established for all affected persons, with a special focus on the health of girls and young women.

## Figures and Tables

**Figure 1 ijerph-20-01050-f001:**
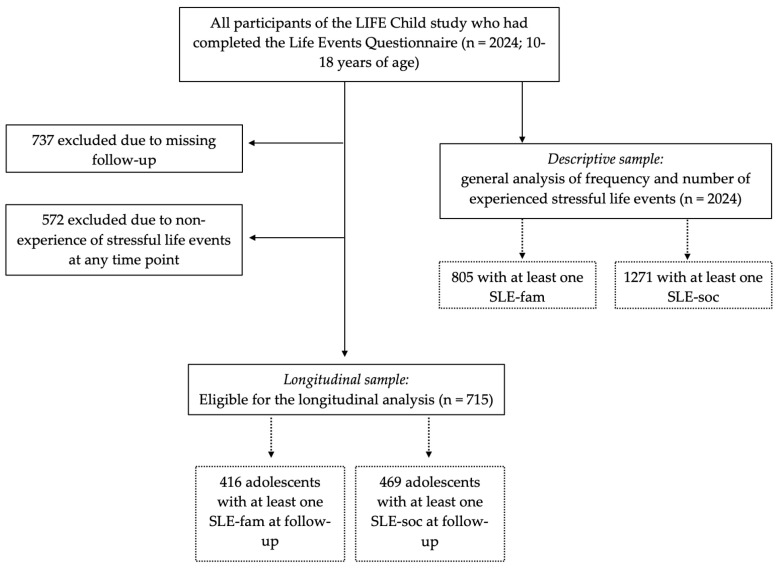
Flow chart of participant selection. SLE-fam: stressful life event in the family domain; SLE-soc: stressful life event in the social environment.

**Figure 2 ijerph-20-01050-f002:**
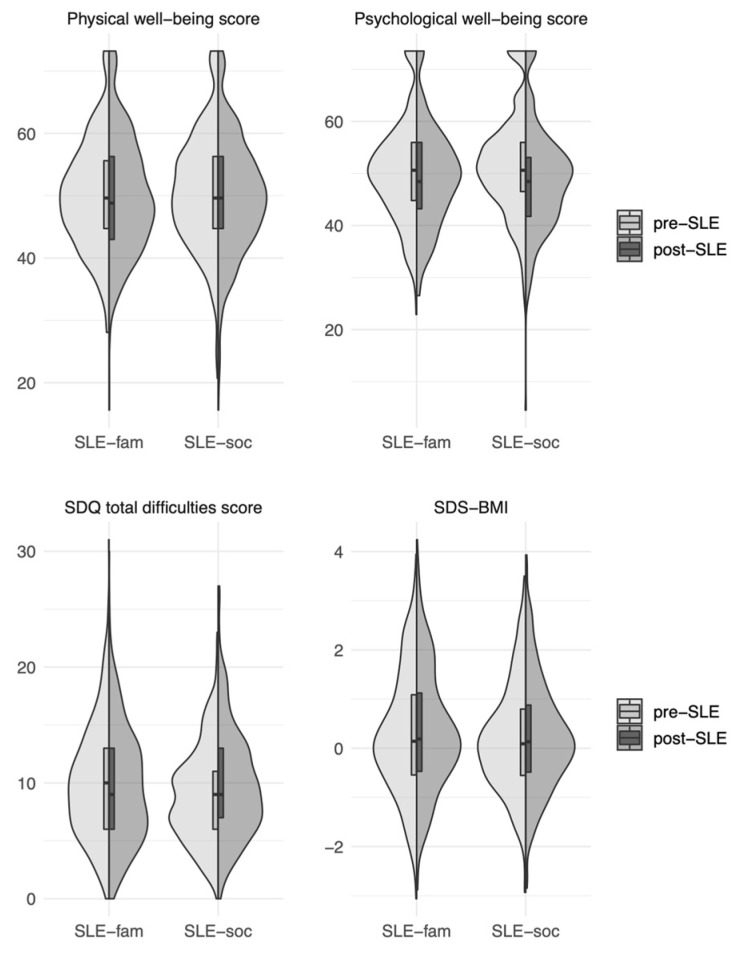
Health parameters before and after an SLE. Distribution of all considered health parameters before and after an SLE had occurred. Score ranges: well-being scores (t-value): 0–100; SDQ total difficulties score: 0–10; BMI-SDS: standard deviation compared to a reference sample. Shaded areas show split violin plots of the density function with inserted boxplots indicating the median and the first and third quartile. pre-SLE = before SLE; post-SLE = after SLE.

**Table 1 ijerph-20-01050-t001:** Frequency of experienced SLE (*n* = 2024).

		Number of Participants, Who Had Experienced This SLE (%)	Number of Participants, Who Had Not Experienced This SLE (%)
**Family problems (SLE-fam)**		805 (39.8)	1205 (59.5)
	Divorce of parents	170 (8.4)	1829 (90.4)
	Death of a close family member	577 (28.5)	1416 (70.0)
	Unemployment within the family	286 (14.1)	1712 (84.6)
**Problems in the social environment** **(SLE-soc)**		1271 (62.8)	712 (35.2)
	Victim of bullying or violence	864 (42.7)	1156 (57.1)
	Problems among friends	616 (30.4)	1390 (68.7)
	Moving to a different place of Residence	249 (12.3)	1751 (86.5)
	Separation from partner	554 (27.4)	1444 (71.3)

Note: an SLE was counted as “experienced” if a participant reported it at least once/at a minimum of one time point. An SLE was counted as “not experienced” if a participant did not report it at any time point.

**Table 2 ijerph-20-01050-t002:** Changes in health parameters from before to after an SLE had occurred.

	KIDSCREEN Physical Well-Being Score	KIDSCREEN Psychological Well-Being Score	SDQ Total Difficulties Score	BMI-SDS
**Family problems (SLE-fam)**				
Number of included participants	414	412	399	416
Pre-SLE-fam	M = 50.06 sd = 8.67	M = 50.50 sd = 9.86	M = 10.02 sd = 4.90	M = 0.28 sd = 1.24
Post-SLE-fam	M = 50.23 sd = 9.45	M = 49.43 sd = 9.81	M = 9.66 sd = 4.96	M = 0.34 sd = 1.24
Difference in means	0.17	−1.07	−0.36	0.06
*p*	0.705	0.02	0.07	<0.01
**Problems in the social environment (SLE-soc)**				
Number of included participants	460	460	438	469
Pre-SLE-soc	M = 50.95 sd = 9.14	M= 51.89 sd = 10.16	M = 8.93 sd = 4.08	M = 0.19 sd = 1.11
Post-SLE-soc	M = 49.78 sd = 9.31	M = 49.14 sd = 10.13	M = 9.92 sd = 4.77	M = 0.23 sd = 1.13
Difference in means	−1.17	−2.74	0.98	0.04
*p*	<0.01	<0.01	<0.01	0.036

Note: Results of paired sample t-tests (longitudinal analysis): pre-SLE = before SLE; post-SLE = after SLE. The differences in means were calculated as follows: mean(post-SLE)-mean(pre-SLE). M = mean, sd = standard deviation, *p* = *p*-value. A total of 416 individuals had at least one SLE-fam after follow-up, and 469 had at least one SLE-soc. Differences are due to missing data in questionnaire responses.

**Table 3 ijerph-20-01050-t003:** Associations between age, sex, and SES and differences in health parameters from before to after an SLE.

Dependent Variable		Independent Variable		
		Age	Female Sex	SES
**Family problems**				
Physical well-being (KIDSCREEN)	b	0.16	−2.53	0.28
	95% CI	−0.29; 0.61	−4.13; −0.92	0.04; 0.51
	*p*	0.49	0.002	0.02
Psychological well-being (KIDSCREEN)	b	0.09	−2.74	0.22
	95% CI	−0.34; 0.52	−4.31; −1.16	−0.01; 0.44
	*p*	0.69	<0.001	0.06
Total difficulties score (SDQ)	b	0.15	0.55	−0.09
	95% CI	−0.08; 0.38	−0.25; 1.35	−0.2; 0.03
	*p*	0.21	0.18	0.15
BMI-SDS	b	−0.0005	0.03	0.003
	95% CI	−0.02; 0.02	−0.04; 0.10	−0.01; 0.01
	*p*	0.96	0.34	0.52
**Problems in the social environment**				
Physical well-being (KIDSCREEN)	b	0.02	−3.94	0.08
	95% CI	−0.36; 0.40	−5.38; −2.49	−0.13; 0.28
	*p*	0.90	<0.0001	0.47
Psychological well-being (KIDSCREEN)	b	−0.08	−4.54	−0.07
	95% CI	−0.52; 0.35	−6.27; −2.80	−0.31; 0.17
	*p*	0.71	<0.0001	0.58
Total difficulties score (SDQ)	b	−0.01	0.58	−0.09
	95% CI	−0.24; 0.21	−0.24; 1.41	−0.21; 0.03
	*p*	0.90	0.17	0.13
BMI-SDS	b	0.002	0.05	−0.01
	95% CI	−0.02; 0.02	−0.02; 0.12	−0.02; 0.003
	*p*	0.83	0.16	0.16

Moderator analysis: b = regression coefficient, 95% CI = confidence interval, *p* = *p*-value. Negative b-values indicate stronger differences in well-being scores and BMI and smaller differences in the SDQ total difficulties score. Positive b-values indicate smaller differences in well-being scores and BMI and stronger differences in the SDQ total difficulties score.

## Data Availability

The legal requirements and the given informed consent do not allow public sharing of the dataset. Data are however available from the authors upon reasonable request and with permission of the research data management of the Medical Faculty, University Leipzig (forschungsdaten@medizin.uni-leipzig.de).

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
