# Peer review of "Associations between Stressful Life Events and Increased Physical and Psychological Health Risks in Adolescents: A Longitudinal Study"

_ijerph, 2023, doi:10.3390/ijerph20021050_

Round 1

Reviewer 1 Report

This study examined the association between stressful life events and several negative health outcomes during adolescence using survey data from the LIFE Child Study in Leipzig, Germany. The manuscript is clearly written and results are presented appropriately. 

My major comments:

1. Although the authors provided in Introduction a summary of previous relevant studies, and mentioned "this paper aims to extend the existing literature with a broad analysis of effects of SLE during adolescence", the novelty of this study is unclear to me, as based on the summary of previous literatures, this research question seems to be substantially studied already.

2. In methods section, the authors need to provide more details on the LIFE Child survey. Particularly, how often do participants fill out questionnaires/come back for a visit? 

3. Related to my 2nd point, the study design is not clear. The authors included participants who completed Life Events Questionnaire between 2011 and 2021, which is a quite long period. If patients filled out questionnaire multiple times during the 10 years, the authors need to state how the index event is determined, e.g., is it the time when participants first responded they experienced a SLE, therefore, can vary from participant to participant, or it's just an universal time point? And, how long are the baseline period (pre index period) and follow up period (post index period) - is it just one year (from index event to the next visit), or the entire period from index to end of study period? The authors also need to mention how "first event" is determined. As the question only captures the previous 6 months, half of time is unobservable. This is a fundamental limitation of the study design. 

4. the final sample should be the 715 participants who meet all the inclusion/exclusion criteria. Descriptive analyses should report based on the final sample, instead of the initial entry cohort of 2024 participants. 

Minor comments:

1. Figure 1 needs borders and arrows to make the flow chart more clear. 

Author Response

1. Although the authors provided in Introduction a summary of previous relevant studies and mentioned "this paper aims to extend the existing literature with a broad analysis of effects of SLE during adolescence", the novelty of this study is unclear to me, as based on the summary of previous literatures, this research question seems to be substantially studied already.

Response: Following the reviewer ́s suggestion, we stated the novelty of our study by including the following sentence:

“Contrasting a time point when adolescents did not report a SLE with a time point after the occurrence of SLE represents the novelty of this study.” (p. 2, line 82-83).

In summary, the uniqueness of or work is the combination of a large sample size (n=2024 for descriptive analysis, n=715 for longitudinal analysis), the broad age range (10-18 years of age), the address of adolescents as a vulnerable population group, the consideration of SLE in various domains of social life, the consideration of different dimensions of psychological and physical health outcomes and the moderation analysis of potential influencing sociodemographic factors. These strengths of the study are described in lines 78-86.

2. In methods section, the authors need to provide more details on the LIFE Child survey. Particularly, how often do participants fill out questionnaires/come back for a visit?

Response: We included the information about the annual follow-up-design of the LIFE Child study, as well as precise details on the follow-up rates of the present sample in the methods section (p. 3, line 106, 118-119).

3. Related to my 2nd point, the study design is not clear. The authors included participants who completed Life Events Questionnaire between 2011 and 2021, which is a quite long period. If patients filled out questionnaire multiple times during the 10 years, the authors need to state how the index event is determined, e.g., is it the time when participants first responded they experienced a SLE, therefore, can vary from participant to participant, or it's just an universal time point? And, how long are the baseline period (pre index period) and follow up period (post index period) - is it just one year (from index event to the next visit), or the entire period from index to end of study period? The authors also need to mention how "first event" is determined. As the question only captures the previous 6 months, half of time is unobservable. This is a fundamental limitation of the study design.

Response: To make our considerations about the two time points "pre-SLE" and "post-SLE" clearer, we added this paragraph to the manuscript:

“In the longitudinal analysis (n = 715), we compared health parameters at two time points, once before experiencing the SLE-fam or SLE-soc, and once after the SLE had occurred. We first identified the time point at which the participant first responded to have experienced a specific SLE within the last six months. Thus, this time point can vary from participant to participant, depending on the individual questionnaire report. This time point was referred to as "post-SLE" (post-SLE-fam or post-SLE-soc). We then determined the time point previous to the "post-SLE"-event, referred to as "pre-SLE" (pre-SLE-fam or pre-SLE-soc). For 92% of participants, the time span between "pre-SLE" and "post-SLE" was one year, meeting the annual follow-up objective of the LIFE Child study. For 7%, the difference was two years, and for 1% it was three or four years. For each subject, only the data points pre-SLE and post-SLE were considered.” (p. 5, line 182 – 194)

We would like to thank the reviewer for remarking this point, as we believe this paragraph improves the clarity and the understandability of the methods significantly.

Regarding the point of criticism of capturing only the previous six months, this limitation is due to the design of the Life Events Questionnaire. We are aware of this disadvantage and have already stated it in the section 5.6, Strengths, limitations. For future study quality enhancements, we will suggest customizing the questions of the life events questionnaire in the LIFE Child study.

4. the final sample should be the 715 participants who meet all the inclusion/exclusion criteria. Descriptive analyses should report based on the final sample, instead of the initial entry cohort of 2024 participants.

Response: In our opinion, the sample of 715 adolescents is too selective for the descriptive analysis. We found it important to investigate the prevalence of SLE in our relatively representative sample of 2024 individuals before selecting those, who had experienced SLE. If we only chose those selected 715 for descriptive analysis, we would not have been able to make a statement about the rates of participants, who had never experienced a SLE (SLE- fam: 59.5 %, SLE-soc: 35.2%). This is important to gain understanding of the burden of SLE straining the LIFE Child participants.

We admit that the distinction between the large and the specific sample was not clear in the first version of the paper. For better understanding, the main sample is now consistently named “descriptive sample” (and the corresponding analyses the descriptive analysis), while the more specific sample/analysis is named “longitudinal sample”/”longitudinal analysis”. This distinction is introduced in the Methods section and used throughout the whole manuscript (including Figure 1). We hope that this makes the differences in samples/analyses clearer.

Minor comments:

1. Figure 1 needs borders and arrows to make the flow chart more clear.

Response: Unfortunately, both figures didn’t display correctly, assumingly due to the PDF converting process during the first submission. Therefore, the reviewers couldn ́t evaluate Figure 1. For submission of the revised manuscript, we are providing the figures in a different data format, hoping that they stay unmodified this time.

Reviewer 2 Report

I appreciated the decision to use a large sample and to carry out a temporally longitudinal study.

I think that the introduction should be enriched by making the reference scientific studies on which the research is based much more explicit.

Citing the bibliographic sources, the Authors should explain why they chose to consider socio-economic status, age and gender as stress moderator variables, neglecting all the others often present in the literature, such as social support.

Individual variables that may play a role in stress mediation, such as coping strategies and social skills, were not considered at all.

In the "Discussions" paragraph, in addition to the participants' subjective perception of the effects of stressful events, the Authors underline that the subjective perception of parents and teachers could also have been collected: I believe that this would have been really necessary, more than possible, because we are dealing with adolescents, that is, subjects who, due to the specificity of their age, can experience mood swings and both amplified and variable perceptions of their own emotional states.

The conclusions are scarce and synthetic and the results should be articulated in more detail.

Author Response

1. I think that the introduction should be enriched by making the reference scientific studies on which the research is based much more explicit.

Response: We gave more details on the reviewed literature, e.g., adding information on sample sizes and ages, country of origin, and main outcomes of the studies (p. 2, line 52-69)

2. Citing the bibliographic sources, the Authors should explain why they chose to consider socio-economic status, age, and gender as stress moderator variables, neglecting all the others often present in the literature, such as social support.
Individual variables that may play a role in stress mediation, such as coping strategies and social skills, were not considered at all.

Response: We added an explanation for our choice of moderating factors into the Introduction:

“Finally, we assessed whether associations between the experience of SLE and health were moderated by the sociodemographic factors age, sex, and SES. Previous studies showed that behavioral difficulties and quality of life of children differ between girls and boys, between different age groups and between different social strata (ref). Therefore, it is to be expected that associations between the experience of SLE and health will also differ between these groups.” (p.2, line 86-92)

We agree that we neglect other important moderator/mediating factors such as social support. This is due to the fact that these characteristics were not assessed in the LIFE Child study. We have stated the neglection of these factors as a study limitation in section 5.6, Strengths, limitations:

“A further limitation is that potential confounding and mediating factors, such as individual social support and coping strategies, were not considered due to limited data availability.” (p.12, line 400-402.)

3. In the "Discussions" paragraph, in addition to the participants' subjective perception of the effects of stressful events, the Authors underline that the subjective perception of parents and teachers could also have been collected: I believe that this would have been really necessary, more than possible, because we are dealing with adolescents, that is, subjects who, due to the specificity of their age, can experience mood swings and both amplified and variable perceptions of their own emotional states.

Response: We share this point of view concerning the missing parent- and teacher- provided reports. Within the LIFE Child study, we are not able to collect reports by teachers. Also, for children aged 11 years and older, we usually refer to reports made by children themselves (instead of parents). We are aware of this limitation and have already mentioned it in the section 5.6 Strengths, limitation.

4. The conclusions are scarce and synthetic and the results should be articulated in more detail.

Response: Following the suggestion of the reviewer, we extended the conclusion by explaining the implications of our research in more detail. We added the following sentence into the conclusion:

“Even in our large healthy adolescent sample from relatively well-situated families, we identified potentially negative health outcomes related to SLE. SLE are therefore of relevance to public health.” (p. 12., line 411-413))

Reviewer 3 Report

I would like to thank the opportunity to review the manuscript titled “Associations between stressful life events and increased physical and psychological health risks in adolescents: a longitudinal study”, I found it very interesting and of relevance for the health sciences. It is astounding the work of the LIFE team and the conclusions of the authors are quite relevant for the clinical field.

I would like the authors to consider the following suggestions for a better comprehension of the article:

In page 3, line 103, I suggest starting figure 1 with capital letter (“Figure 1”). Also, I have problems to see the flow chart, with not any line or box joining the text. I suggest revising this question, as Figure 1 cannot be properly understood.

Regarding measurements, it would enhance the quality of the article to add the reliability of each scale and questionnaire, also in the Life Events Questionnaire if they could. Also, data from the validation of the scales to the German context (KIDSCREEN-27 and SDQ mostly).

Finally, I suggest reviewing the References section, as it does not fulfill entirely the requirements (bold font, italics…).

Author Response

1. In page 3, line 103, I suggest starting figure 1 with capital letter (“Figure 1”). Also, I have problems to see the flow chart, with not any line or box joining the text. I suggest revising this question, as Figure 1 cannot be properly understood.

Response: We changed “figure 1” to “Figure 1”.
Unfortunately, both figures didn’t display correctly, assumingly due to the PDF converting process during the first submission. Therefore, the reviewers could not evaluate Figure 1. For submission of the revised manuscript we are providing the figures in a different data format, hoping that they stay unmodified this time.

2. Regarding measurements, it would enhance the quality of the article to add the reliability of each scale and questionnaire, also in the Life Events Questionnaire if they could. Also, data from the validation of the scales to the German context (KIDSCREEN-27 and SDQ mostly).

Response: We added the reliability of the SDQ total difficulties score, as well as the physical- and psychological- wellbeing scores of the KIDSCREEN-27, using the internal consistency calculation method “Cronbach ́s alpha” (see section 2.3 and 2.4).
All scores showed satisfactory internal consistencies. We compared our internal consistency findings to reference populations, finding no significant difference. Furthermore, we added the validation studies of both instruments to the list of references.

For the Life Events Questionnaire, the calculation of internal consistency was not applicable as the items refer to different events (they do not represent items on the same scale).

3. Finally, I suggest reviewing the References section, as it does not fulfill entirely the requirements (bold font, italics...).

Response: the reference section has been reviewed.

Reviewer 4 Report

Thank you for the opportunity to revise this interesting research. Authors have analyzed data from a longitudinal investigation to study associations between stressful life events  and negative health outcomes in adolescents aged 10-18 (n=2024). Moderating effects of sociodemographic factors were tested.  

Overall, the study is very well organized and comprehensively described, and is expected to have a significant contribution to our knowledge on adolescent health.

However, this reviewer has some concerns and comments:

1-    Contextual information are much needed. This study was conducted in one German city only. It is very important for readers and researchers to understand the context of this city and characteristics of its population. This can also help us gauge the generalizability of the findings.

2-    The introduction seems to focus on the uniqueness of this study as being a longitudinal investigation. Critiques to previous research were also made mainly based on their cross-sectional nature. However, the two time assessment points really limit the implications of the study. I was not even sure why the six months cutoff score was used in this way. Couldn’t it be possible to simply ask the participant on the exact time of the SLE? At least so we do not falsely record those who experienced major SLEs within 7 months or more.

3-    There is no mentioning of how the moderation analysis was conducted and the theoretical framework of the method selection.

4-    Did the author consider a mediation analysis? What factors could possibly function as a go-between SLE and adolescent health outcomes?

5-    Please include the psychometrics of the utilized scales (past studies as well as current one).

6-    Minor: Figure 1 does not display correctly

Thank you for the great efforts!

Author Response

1. Contextual information are much needed. This study was conducted in one German city only. It is very important for readers and researchers to understand the context of this city and characteristics of its population. This can also help us gauge the generalizability of the findings.

Response: Following the suggestion of the reviewer, we included a paragraph containing contextual information on the population and socioeconomic structure of Leipzig:

“Participants mainly come from the urban and rural areas of Leipzig, a city in Eastern Germany with more than 600,000 inhabitants (as for 2021). Compared to other, especially West German cities, Leipzig is a city with a relatively high poverty and unemployment rate. Nevertheless, Leipzig has been reported to be a city with a high living quality.” (p. 3, line 106-110)

2. The introduction seems to focus on the uniqueness of this study as being a longitudinal investigation. Critiques to previous research were also made mainly based on their cross- sectional nature. However, the two time assessment points really limit the implications of the study. I was not even sure why the six months cutoff score was used in this way. Couldn’t it be possible to simply ask the participant on the exact time of the SLE? At least so we do not falsely record those who experienced major SLEs within 7 months or more.

Response: In our opinion, the great novelty of our study is the comparison of health parameters at two time points, once before and once after the occurrence of a SLE. We have made this more clear by including the following sentence into the introduction:

“Contrasting a time point when adolescents did not report a SLE with a time point after the occurrence of SLE represents the novelty of this study.” (p. 2, line 82-83)

It is true that we follow a short time longitudinal study design, covering only two time points. Therefore we cannot make statements about the long-term effects of SLE, as we have already stated in the section 5.6 Strengths, limitations. Nevertheless, we believe that we can make more precise statements on effects of SLE this way than using a simple cross-sectional design.

Regarding the point of criticism of capturing only the previous six months, this limitation is due to the design of the Life Events Questionnaire. We are aware of this disadvantage and have already stated it in the section 5.6, Strengths, limitations. For future study quality enhancements, we will suggest customizing the questions of the Life Events Questionnaire in the LIFE Child study.

3. There is no mentioning of how the moderation analysis was conducted and the theoretical framework of the method selection.

Response: We described the analysis background of the moderator analysis in section 2.7, Data analysis. We made this clearer by including an explanation in brackets (p. 5, line 201). The moderation analysis was examined using multiple linear regression models.

4. Did the author consider a mediation analysis? What factors could possibly function as a go- between SLE and adolescent health outcomes?

Response: We thank the reviewer for this comment. Indeed, we discussed potential mediating factors that may influence the relationship between SLE and health disadvantages. For instance, one may speculate that divorce of the parents accompanies impaired social support within the parental home, potentially leading to lower psychological well-being. Furthermore, individual coping skills may play a role mediating these associations.

For our study, we focused mainly on the moderator analysis, as sociodemographic characteristics are of great relevance for the understanding of risk factors threatening adolescents’ health. Furthermore, our data set lacks applicable data for a mediating analysis. For example, coping strategies and social support are not assessed in this study.

We have added the following sentence into the section 5.6 Strengths, Limitations:

“A further limitation is that potential confounding and mediating factors, such as individual social support and coping strategies, were not considered due to limited data availability.” (p. 12, line 400-402.)

5. Please include the psychometrics of the utilized scales (past studies as well as current one).

Response: We added the reliability of the SDQ total difficulties score, as well as the physical- and psychological- wellbeing scores of the KIDSCREEN-27, using the internal consistency calculation method “Cronbach ́s alpha” (see section 2.3 and 2.4).
All scores showed satisfactory internal consistencies. We compared our internal consistency findings to reference populations, finding no significant difference. Furthermore, we added the validation studies of both instruments to the list of references.

6. Minor: Figure 1 does not display correctly

Response: Unfortunately, both figures didn’t display correctly, assumingly due to the PDF converting process during the first submission. Therefore, the reviewers could not evaluate Figure 1. For submission of the revised manuscript, we are providing the figures in a different data format, hoping that they stay unmodified this time.

Round 2

Reviewer 1 Report

The authors have appropriately addressed my comments. 

Reviewer 2 Report

I appreciated the effort in increasing the details, improving the quality of presentation and increasing the references to literature. Thank You

Reviewer 4 Report

Many thanks to the authors for the changes made to the manuscript. The paper has been improved and suggestions have been adequately considered.